# Transcriptomic Analysis Reveals Host miRNAs Correlated with Immune Gene Dysregulation during Fatal Disease Progression in the Ebola Virus Cynomolgus Macaque Disease Model

**DOI:** 10.3390/microorganisms9030665

**Published:** 2021-03-23

**Authors:** Christopher P. Stefan, Catherine E. Arnold, Charles J. Shoemaker, Elizabeth E. Zumbrun, Louis A. Altamura, Christina E. Douglas, Cheryl L. Taylor-Howell, Amanda S. Graham, Korey L. Delp, Candace D. Blancett, Keersten M. Ricks, Scott P. Olschner, Joshua D. Shamblin, Suzanne E. Wollen, Justine M. Zelko, Holly A. Bloomfield, Thomas R. Sprague, Heather L. Esham, Timothy D. Minogue

**Affiliations:** 1Diagnostic Systems Division, U.S. Army Medical Research Institute of Infectious Diseases, Ft. Detrick, MD 21702, USA; catherine.e.arnold13.civ@mail.mil (C.E.A.); charles.j.shoemaker5.ctr@mail.mil (C.J.S.); louis.altamura@gmail.com (L.A.A.); christina.e.burrows.ctr@mail.mil (C.E.D.); cheryl.l.taylor-howell.ctr@mail.mil (C.L.T.-H.); Amanda.Graham@ST.DHS.GOV (A.S.G.); korey.l.delp.ctr@mail.mil (K.L.D.); candace.d.blancett.ctr@mail.mil (C.D.B.); keersten.m.ricks.civ@mail.mil (K.M.R.); scott.p.olschner.civ@mail.mil (S.P.O.); 2Virology Division, U.S. Army Medical Research Institute of Infectious Diseases, Ft. Detrick, MD 21702, USA; elizabeth.e.zumbrun.civ@mail.mil (E.E.Z.); joshua.d.shamblin1.civ@mail.mil (J.D.S.); swollen-roberts@hivresearch.org (S.E.W.); Justinezelko@gmail.com (J.M.Z.); holly.a.bloomfield.civ@mail.mil (H.A.B.); thomassprague3@yahoo.com (T.R.S.); heather.l.esham.civ@mail.mil (H.L.E.)

**Keywords:** Ebola virus, non-human primate, pathogenesis, transcriptome, immunology, miRNA, mRNA, cytokine, inflammation, apoptosis

## Abstract

Ebola virus is a continuing threat to human populations, causing a virulent hemorrhagic fever disease characterized by dysregulation of both the innate and adaptive host immune responses. Severe cases are distinguished by an early, elevated pro-inflammatory response followed by a pronounced lymphopenia with B and T cells unable to mount an effective anti-viral response. The precise mechanisms underlying the dysregulation of the host immune system are poorly understood. In recent years, focus on host-derived miRNAs showed these molecules to play an important role in the host gene regulation arsenal. Here, we describe an investigation of RNA biomarkers in the fatal Ebola virus disease (EVD) cynomolgus macaque model. We monitored both host mRNA and miRNA responses in whole blood longitudinally over the disease course in these non-human primates (NHPs). Analysis of the interactions between these classes of RNAs revealed several miRNA markers significantly correlated with downregulation of genes; specifically, the analysis revealed those involved in dysregulated immune pathways associated with EVD. In particular, we noted strong interactions between the miRNAs hsa-miR-122-5p and hsa-miR-125b-5p with immunological genes regulating both B and T-cell activation. This promising set of biomarkers will be useful in future studies of severe EVD pathogenesis in both NHPs and humans and may serve as potential prognostic targets.

## 1. Introduction

Ebola virus is a member of the negative single strand RNA virus family *Filoviridae*, noted for causing a virulent hemorrhagic fever in humans and non-human primates (NHPs) termed Ebola virus disease (EVD). The genus *Ebolavirus* includes five species with case fatality rates up to 90%. *Zaire Ebolavirus* (EBOV) is the most common species in human outbreaks and has claimed the most lives to date. Natural outbreaks of filoviruses in humans have occurred in numerous countries in Central and Western Africa since its discovery in 1976, with the 2013–2016 EBOV outbreak in West Africa being the largest to date. Characteristic EVD symptoms include fever, myalgia, headache, and gastrointestinal symptoms with patients potentially developing a maculopapular rash [1,2]. Fatal outcomes correlate with increased viremia, impaired immune responses, septic shock, disseminated intravascular coagulation, and multi-organ failure [1,2,3]. Promising EVD experimental vaccines and therapeutics have emerged in recent years; however, the primary treatment regimen for infected persons is intensive supportive care focused on maintenance of fluids, electrolytes, and blood pressure [4]. Early diagnosis of EVD is critical for implementation of effective clinical interventions as well as limiting disease spread.

In general, both NHP and human survivors of hemorrhagic fever viruses control viral loads early during the infection. In these cases, balanced immune responses are characterized by modest and transient early pro-inflammatory responses that transition to virus-directed antibody and cell-mediated adaptive responses [5,6]. By contrast, severe or fatal outcomes are associated with high viremia and dysregulated immune responses, notably a significant early pro-inflammatory reaction. This hyperactive immune state is thought to be driven by selective EBOV infection of macrophages and immature monocytes, which in turn secrete key cytokine/chemokine mediators of inflammation [7,8,9]. Paired with this exaggerated inflammatory response is a greatly diminished effective B and T-cell response. For example, in humans, a robust IgG antibody response to EBOV antigens is largely absent or diminished in fatal cases [5]. In severe EVD cases, the initial pro-inflammatory immune response manifests at both the gene and protein levels. This inflammatory state has numerous secondary effects, including the premature activation of immune pathways associated with the maturation and activation of B cells, cytotoxic T cells, and Natural Killer (NK) cells [6,10,11,12]. This early activation is later followed by widespread lymphocyte apoptosis [13]. The precise mechanism leading to this lymphopenic end state and how it relates to the initial viral-driven inflammation response remain unresolved.

An emerging field in diagnostics involves host transcriptomic biomarkers, which includes the total RNA changes present in various relevant clinical matrices compared to uninfected baseline. One of the more relevant clinical matrices for this analysis is whole blood as it is an easily accessible systemic tissue. The presence in blood of multiple cell types relevant to immune response, including lymphocytes, macrophages, monocytes, eosinophils, basophils, and neutrophils, allows for the surveillance of diverse transcriptomic biomarkers during the course of infection, including messenger RNA (mRNA) and microRNA (miRNA) [14]. Our lab and many others have reported mRNA transcriptomic responses to EBOV infection in NHPs [15,16,17,18]. Early mRNA biomarkers that have been identified and which correlate with human EVD are largely players in the innate immune response pathway; examples of such genes include *DDX58, IFI44, IFIT2, IFIT3, MX1*, and *OASL*.

While host mRNAs are common targets for pathogen signature discovery, miRNAs are emerging as appealing diagnostic targets. miRNAs are short (~22 nt) non-coding RNA sequences that control gene expression through binding specific mRNAs, consequently targeting them for degradation and limiting their translation [19]. miRNAs can be of either host or viral origin; examples of virally derived miRNAs can be found with viruses as diverse as Epstein-Barr Virus and EBOV [20]. In the instance of EBOV, multiple virus-encoded miRNAs have been discovered, including EBOV-miR-1-5p, which is believed to play a role in evasion of the host immune response [21]. The vast majority of detectable miRNAs, however, are host-derived and these play important roles in the maintenance of homeostasis, both under healthy conditions and as a response to microbial infection. In comparison to mRNAs or proteins, fewer discrete miRNAs have been identified as each miRNA can regulate multiple gene expression networks [22]. While mostly studied as markers of cancer biogenesis, miRNAs have been recognized in recent years as important regulators of inflammatory and immune processes linked with infectious diseases including viral infection [23,24,25,26]. Limited research has been reported on the role that host-derived miRNAs play in modulating the immune response surrounding EBOV infection. We chose to investigate their role in Cynomolgus macaques (*Macaca fascicularis*) during EVD; this NHP model for EBOV infection is favored due to the similarities in disease course seen relative to humans [27]. In this study, we report a high-resolution temporal analysis, including during pre-symptomatic timepoints, of both host mRNA and miRNA transcriptomic responses in cynomolgus macaques during fatal EBOV infection. Furthermore, we identify potential miRNA regulation of key immune pathways that are implicated in severe EVD.

## 2. Materials and Methods

### 2.1. Ethics Statement

This work was supported by an approved USAMRIID Institutional Animal Care and Use Committee (IACUC) animal research protocol (AP-17-020; approved September 2017) in compliance with the Animal Welfare Act, PHS Policy, and other Federal statutes and regulations relating to animals and experiments involving animals. The facility where this research was conducted is accredited by the Association for Assessment and Accreditation of Laboratory Animal Care, International and adheres to principles stated in the Guide for the Care and Use of Laboratory Animals, National Research Council, 2011. Approved United States Army Medical Research Institute of Infectious Diseases (USAMRIID) animal research protocols undergo an annual review every year. Animals are cared for by a large staff of highly qualified veterinarians, veterinary technicians, and animal caretakers. All personnel caring for and working with animals at USAMRIID have substantial training to ensure only the highest quality animal care and use. All steps were taken to enrich the welfare and to avoid the suffering of the animals in accordance with the “Weatherall report for the use of nonhuman primates” recommendations. Animals were housed in adjoining individual primate cages allowing social interactions, under controlled conditions of humidity, temperature, and light (12 h light/12 h dark cycles). Food and water were available ad libitum. Animals were monitored and fed commercial monkey chow, treats, and fruit twice daily by trained personnel. Environmental enrichment consisted of commercial toys. Post-exposure, animals were evaluated daily for signs of illness. Following development of clinical signs, animals were checked multiple times daily. Institute scoring criteria were used to determine timing of humane euthanasia under anesthesia. Highly trained personnel completed all procedures under the oversight of an attending veterinarian and all invasive clinical procedures were performed while animals were anesthetized. NHPs were humanely euthanized by administration of greater than or equal to 6 mg/kg Telazol until a surgical plane of anesthesia was achieved, terminally bled via intracardiaccly (IC) puncture, and administered 0.3–0.4 ml/kg pentobarbital-based euthanasia solution (Fatal-Plus) IC.

### 2.2. Animals and Study Design

Nine, clinically normal, adult cynomolgus macaques (*Macaca fascicularis*), originally obtained from World Wide Primates and housed in the USAMRIID NHP colony, were used for this study. The NHPs, mixed male (*n* = 5) and female (*n* = 4), were 8 to 9 years of age, and ranged in body weight from 4.80 to 10.40 kg at the time of challenge. None of these NHPs were exposed to infectious pathogens in previous studies, and all were serologically screened by ELISA and found to be negative for filovirus (Marburg/Ebola), Herpes B, STLV-1, SIV, SRV1, 2, and 3, Tuberculosis, *Salmonella*, *Campylobacter*, hypermucoid HVM *Klebsiella*, and *Shigella* infections. For the EBOV challenge, eight animals were challenged intramuscularly in the right lateral thigh with 1 mL of prepared virus inoculum with a target dose of 100 plaque forming units (PFU). The actual challenge dose was later determined to be 152.5 PFU as determined by plaque assay. For an uninfected control, one animal was injected intramuscularly (in the thigh) with 1 mL of diluent consisting of Minimum Essential Media (MEM) +2% Heat Inactivated Fetal Calf Serum (HI-FCS). After EBOV exposure, all animals were monitored for temperature changes (rectal checks), while weight loss trends, clinical signs of disease, survival trends, and blood samples were collected on days −14, −7, and 1–8 for virological, molecular, and chemical analyses. Animals were humanely euthanized when they reached a predetermined euthanasia score of 4 (out of 4). The uninfected control animal was euthanized immediately after the first EBOV challenged animal was euthanized at 6 days post-infection (DPI). All experiments using live virus were performed in an animal Biosafety Level 4 (ABSL-4) containment laboratory at USAMRIID with trained personnel in positive pressure encapsulating suits.

### 2.3. Challenge Virus Stock

EBOV strain Kikwit95-7U was originally isolated from a 65 year female during an outbreak occurring in 1995 in the Democratic Republic of the Congo (formerly Zaire). The patient exhibited disease, was hospitalized, and died. The passage of virus, designated virus seed pool 807223, was conducted at the CDC using Vero E6 cells. A second passage of virus, designated WRC000121, was conducted at the University of Texas Medical Branch. WRC000121 was transferred to USAMRIID and propagated on BEI-Vero E6 cells to produce the USAMRIID master seed stock, R4542, a 7U variant. The stock was subsequently plaque titrated, sequenced, and confirmed negative for mycoplasma and endotoxin. On the day of challenge, viral inoculum was prepared using viral seed stock diluted with Minimum Essential Media (MEM) + 2% Heat Inactivated Fetal Calf Serum (HI-FCS). The challenge dose received by the animals was 152.5 PFU as determined by neutral red plaque assay as previously described [28].

### 2.4. Blood Chemistries and Sample Collection

Serum was isolated using a gel-based serum separator (Sarstedt, Numbrecht, Germany) and was stored at −80 °C for subsequent analysis. Blood chemistries were performed on serum with a Piccolo chemistry analyzer (Abaxis, Union City, CA, USA) utilizing General Chemistry 13 detection discs according to manufacturer’s instructions. Whole blood, saliva, nasal secretions, and rectal swabs were collected from anesthetized animals. Urine was collected from free-catch metabolic pans. All whole blood samples were initially diluted 1:1 with purified molecular grade water, prior to being vigorously mixed with a 3:1 ratio of TRIzol™ LS (ThermoFisher Scientific, Waltham, MA, USA). All other sample types were mixed at a 3:1 TRIzol™ LS ratio and were then frozen at −80 °C until subsequent extraction.

### 2.5. EBOV Antibody and Antigen Detection

For EBOV antibody detection, we utilized recombinant EBOV antigens produced in Drosophila S2 cell lines by Axel Lehrer at the University of Hawaii; these included glycoprotein (GP), nucleoprotein (NP), and viral matrix protein (VP40). All antigen targets were conjugated to magnetic microspheres using the Luminex xMAP^®^ antibody coupling kit (Luminex Inc., Austin, TX, USA) according to the manufacturer’s instructions. Briefly, 500 µL of Magplex microspheres (12.5 × 10^6^ microspheres/mL) were washed three times using a magnetic microcentrifuge tube holder, and resuspended in 480 µL of activation buffer. Then, 10 µL of both sulfo-*N*-hydroxysulfosuccinimide (sulfo-NHS) and 1-Ethyl-3-(3-dimethylaminopropyl) carbodiimide hydrochloride (EDC) solutions were added to the resuspended microspheres. The tube was covered with aluminum foil and placed on a benchtop rotating mixer for 20 min. After surface activation with EDC, the microspheres were washed three times with activation buffer prior to adding the recombinant protein antigen at a final concentration of 4 µg antigen: 1×10^6^ microspheres. This concentration of recombinant protein coupled to the surface of microspheres has been shown to be optimal for IgG and IgM detection. The tube was again covered with aluminum foil and placed on a benchtop rotating mixer for 2 h. After this coupling step, the microspheres were washed three times with wash buffer and resuspended in 500 µL of wash buffer for future use. EBOV NP, VP40, and GP were coupled to Magplex microsphere regions #66, #77, and #19, respectively, in order to facilitate multiplexing experiments. Beads were stored at 4 °C until further use.

NHP serum samples were diluted 1:100 in 1X PBS containing 0.02% Tween-20 (Sigma, St. Louis, MO, USA) (PBST) and 5% skim milk (PBST-SK). Each individual antigen-coupled bead was mixed at a 1:1 ratio prior to diluting in PBST to 5×10^4^ microspheres/mL, and the mixture was added to the wells of a Costar polystyrene 96-well plate at 50 µL per well (2500 microspheres of each antigen bead set/well). The plate was placed on a magnetic plate separator (Luminex Corp, Austin, TX, USA), covered with foil, and microspheres were allowed to collect for 60 s. While still attached to the magnet, the buffer was removed from the plate by inverting. Then, 50 µL of diluted serum samples were added to appropriate wells. The plate was covered with a black, vinyl plate cover and incubated with shaking for 1 h at ambient temperature. The plate was washed three times with 100 µL of PBST for each wash, using the plate magnet to retain the Magplex microspheres in the wells. Liquid was discarded by inverting as above. Next, 50 µL of a 1:100 dilution of mouse anti-human IgM phycoerythrin conjugate (Invitrogen, MA1-10381, Carlsbad, CA, USA) or goat anti-human IgG phycoerythrin conjugate (Sigma, P9170, St. Louis, MO, USA) in PBST-SK was added to the appropriate wells. The plate was covered again with a black, vinyl plate sealer and incubated with shaking for 1 h at ambient temperature. After incubation, the plate was washed three times as detailed above, and the Magplex microspheres were resuspended in 100 µL of PBST for analysis on the Magpix instrument (Luminex Corp, Austin, TX, USA). Raw data was reported as median fluorescence intensity for each bead set in the multiplex.

For EBOV antigen detection, anti-VP40 (A10A produced at USAMRIID), anti-GP (AH05 produced at USAMRIID), and anti-NP (ZNP31-1-8 produced by Hokkaido University Research Center) monoclonal antibodies were coupled to Magplex magnetic microspheres as described above. Anti-VP40, anti-GP, and anti-NP monoclonal antibodies were coupled to bead regions #77, #19, and #26, respectively. Anti-VP40 (AE11 produced at USAMRIID), anti-GP (13C6 produced at USAMRIID), and anti-NP (ZNP24-4-2 produced by Hokkaido University Research Center) detector monoclonal antibodies were biotinylated per EZ-Link™ Sulfo-NHS-Biotin kit instructions (Thermo Scientific, Waltham, MA, USA). NHP plasma samples were diluted 1:20 in 1X PBS containing 0.02% Tween-20 (Sigma, St. Louis, MO, USA) (PBST) and 5% skim milk (PBST-SK). Each individual antigen-coupled bead was mixed at a 1:1 ratio prior to diluting in PBST to 5×10^4^ microspheres/mL, and the mixture was added to the wells of a Costar polystyrene 96-well plate at 50 µL per well (2500 microspheres of each antigen bead set/well). The plate was placed on a magnetic plate separator (Luminex Corp, Austin, TX, USA), covered with foil, and microspheres were allowed to collect for 60 sec. While still attached to the magnet, the buffer was removed from the plate by inverting. Then, 50 µL of diluted serum samples were added to appropriate wells. The plate was covered with a black, vinyl plate cover and incubated with shaking for 1 h at ambient temperature. The plate was washed three times with 100 µL of PBST for each wash, using the plate magnet to retain the Magplex microspheres in the wells. Liquid was discarded by inverting as above. Next, 50 µL of a 4 µg/mL solution of all three biotinylated detector antibodies diluted in PBST-SK was added to the wells. The plate was covered again with a black, vinyl plate sealer and incubated with shaking for 1 h at ambient temperature. After incubation, the plate was washed three times as detailed above prior to adding 50µL of streptavidin-phycoerythrin (SAPE, ThermoFisher Scientific, Waltham, MA, USA) diluted in PBST-SK, and incubated for 30 min at RT, covered with a black plate sealer. Finally, the plate was washed three times with 100 µL of PBST. The Magplex microspheres were resuspended in 100 µL of PBST for analysis on the Magpix instrument (Luminex Corp, Austin, TX, USA). Raw data were reported as median fluorescence intensity for each bead set in the multiplex.

### 2.6. RNA Extraction and Quantitation

RNA for RT-qPCR analysis was extracted with the Qiagen EZ1 Advanced robot with the EZ1 Virus Mini 2.0 Kit (Qiagen, Germantown, MD, USA) according to the manufacturer’s directions. RNA for NanoString and miRNA sequencing analysis was manually extracted using the miRNeasy kit (Qiagen, Germantown, MD, USA) following the manufacturer’s instructions with the modification that the initial TRIzol™ LS/water/whole blood mixture was placed in a Phasemaker phase separation tube (Invitrogen) prior to being centrifuged at 12,000× *g* for 15 min at room temperature. The top aqueous layer was then transferred into a 2 mL tube with 1.5 times the volume of 100% ethanol added. The samples were then transferred to a miRNeasy kit spin column 700 μL at a time. Each sample was centrifuged for 30 s at 8000× *g* until the entire sample passed through the column. Next, the samples were cleaned by adding 700 μL of buffer RWT to each column and centrifuged for 30 s at 8000× *g*, followed by 500 μL of buffer RPE being added to the columns and centrifuged twice for 30 s at 8000× *g*. A final drying spin at 10,000× *g* for 2 min removed any residual ethanol from the columns. The samples were then eluted in 50 μL RNase-free water by centrifugation for 2 min at 10,000× *g*. Purified RNA samples were quantitated using the Qubit RNA HS Assay Kit (ThermoFisher Scientific, Waltham, MA, USA) according to the manufacturer’s recommendations and concentrations were reported in ng/μL.

### 2.7. Determination of Viremia in NHP Clinical Specimens by RT-qPCR

For all RT-qPCR assays, the previously designed Ebola-Zaire MGB assay targeting the viral NP was utilized [29]. Standard curves using RNA generated from EBOV Kikwit virus were run for each sample matrix, and both positive control compensation wells and internal negative control wells were run on each plate. Reactions were run on the LightCycler 480 (Roche, Basel, Switzerland) with the following cycling conditions: 50 °C for 15 min (1 cycle); 95 °C for 5 min (1 cycle); 95 °C for 1 s and 60 °C for 20 s (45 cycles); and 40 °C for 30 s (1 cycle). A single fluorescence read was taken at the end of each 60 °C step, and a sample was considered positive if the Cq value was less than 40 cycles. Samples were assayed in technical triplicate.

### 2.8. NanoString Processing and Analysis

We performed targeted gene expression profiling using total RNA isolated from whole blood using the NanoString NHP Immunology v. 2 panel CodeSet (NanoString Technologies, Inc., Seattle, WA, USA). Sample preparation for NanoString nCounter assays was performed according to the manufacturer’s protocols. In brief, 8 μL of Reporter CodeSet was combined with 2 μL of Capture ProbeSet and 5 μL sample RNA. We diluted RNA samples as necessary for a total concentration of 50 ng in 5 μL. In cases where the RNA concentration was insufficient, we used 5 μL of the undiluted samples. The reactions were incubated for 17 h at 65 °C, followed by a 4 °C hold step until ready for analysis. After the hybridization period, the samples were loaded on NanoString SPRINT cartridges. The cartridges were then imaged by a SPRINT Profiler System (NanoString Technologies, Inc., Seattle, WA, USA), which counts the fluorescent molecular barcodes present in the cartridge. This data is converted to a text file (RCC) which indicates total counts per gene target (i.e., gene expression).

All RCC files were imported into the nSolver 4.0 software to perform the following: assay quality assessments, background determination, positive control spike-in normalization, and reference gene normalization. The nCounter Advanced Analysis Module (nCAAM; v. 2.0.115) was used to perform initial quality control assessments, reference gene identification, and other analyses. Quality control (QC) was performed on RCC files, samples that did not pass initial QC were re-run on the NanoString nSolver platform. Raw counts were then normalized to geometric mean counts of the synthetic positive controls included on the codeset. nCAAM and the geNorm algorithm was used to normalize each dataset by taking the five most stable housekeeping genes across samples and using it on all datasets (Appendix A). Housekeeping genes used are listed in Appendix A. For secondary analysis, data were processed using JMP genomics 8.1 software (SAS Institute, Cary, NC, USA). For sample binning, samples were binned into four phases including incubation, early, mid, and late phases. Samples were categorized as follows: incubation phase samples were infected but not viremic, early phase samples were viremic and non-symptomatic, mid phase samples were viremic with responsiveness scores <2, and late phase samples were viremic with responsiveness scores >2. Expression changes for each phase were compared to day −7 and −14 non-infected control samples. Variability resulting from individual NHPs was removed prior to analysis. Cutoff for differential gene expression was ±1 log2 fold-change with an FDR-adjusted *p*-value of 0.05.

### 2.9. miRNA Library Preparation, Sequencing and Analysis

Individual miRNA sequences were assigned Unique Molecular Indexes (UMI) using QIAseq miRNA Library Prep kit (Qiagen, Germantown, MD, USA) with QIAseq miRNA NGS 48 Index IL adapters (Qiagen, Germantown, MD, USA) following manufacture recommendations. Sequencing libraries were quantified using a Qubit Fluorometer (ThermoFisher Scientific, Waltham MA, USA) and library size was determined using the Agilent Tapestation HS DNA 1000 assay (Agilent technologies, Santa Clara, CA, USA). Libraries where normalized to 2 nM and quantified using KAPA Biosystems Library Quantification Kit for Illumina platforms (KAPA Biosystems, Boston, MA, USA). Libraries were pooled at equal volume ratios. Pooled libraries were denatured with 0.1N NaOH for 5 min then further diluted to 10–14 pM prior to sequencing on the MiSeq System (Illumina, San Diego, CA, USA) using a MiSeq kit V3 150 cycle kit following manufacture recommendations.

Primary analysis on raw FASTQ data, including mapping miRNA sequences and counting UMIs, was performed using Qiagen Gene Globe Data Analysis Center (Appendix A) (Qiagen, Germantown, MD, USA). For secondary analysis, data were processed using JMP genomics 8.1 software (SAS Institute, Cary, NC, USA). Counts were given a threshold of 10 and normalized to counts per million reads based on total counts for each sample. Data were log2 transformed prior to further analysis. Comparison of days -7 and -14 DPI resulted in one significant differentially expressed miRNA/piRNAs (DEM), hsa-miR-6126; therefore, these datasets were combined and utilized as pairwise comparators to all DPI similar to mRNA analysis. One pre-infection NHP sample was identified as an outlier and removed prior to analysis. Variability resulting from individual NHPs was removed prior to analysis. Cutoff for differential gene expression was ±1 log2 fold-change with an FDR-adjusted *p*-value of 0.05.

### 2.10. Pathway and Correlation Analysis

Ingenuity Pathway Analysis (IPA) (Qiagen) was used to determine enriched canonical pathways and predicted upstream regulators using the comparison analysis. CIBERSORT was used with averaged normalized mRNA counts for each day to determine estimated cell population quantities using the LM22 human mixture dataset for comparison with 500 permutations for statistical analysis [30]. KEGG pathway analysis was performed on differentially expressed miRNAs using DIANA-miRPath v3 [31]. Predicted interactions between miRNAs and mRNAs were discovered using several databases within IPA including TarBase, TargetScan, miRecords, and Ingenuity expert findings. Spearman Rho correlations were calculated using JMP genomics 8.1 on matched NHP miRNA and mRNA normalized counts using DPI 0–8.

## 3. Results

### 3.1. Challenge Design and Sample Characterization

Intramuscular challenge of eight NHPs with a 100 PFU target dose (152.5 actual dose) of Ebola-Kikwit with one sham infected (MEM + 2% HI-FCS diluent), showed that all infected NHPs developed clinical signs starting at 5 days post-infection (DPI). Infected NHPs progressed to a severe disease state and reached endpoint euthanasia criteria between 6–8 DPI (Figure 1a,b). All blood samples, which were collected terminally, showed elevated levels of the liver enzymes aspartate aminotransferase (AST) and alanine aminotransferase (ALT), which are blood chemical markers of liver distress common in primates exposed to EBOV (Appendix A) [32,33]. Additionally, we observed elevated levels of blood urea nitrogen (BUN) and serum creatinine (CRE), which are markers of renal damage [34]. Other clinical signs of EVD in NHPs included widespread maculopapular rash (8/8 infected animals), lymphadenopathy (2/8 infected animals), and nasal bleeding (1/8). No clinical signs of disease were observed in the uninfected control animal at any time during the study.

CIBERSORT, a digital cell sorting program, determined the immune cell populations present during the course of infection in lieu of a complete blood count (CBC). Specifically, normalized counts for the NanoString NHP Immunology v. 2 codeset served as an input yielding predicted relative cell population numbers for nine different immune cell populations (Figure 1c). This transcriptomic signature indicated that NHPs experienced loss of lymphocytes, including both T- and B-cells, during disease progression. Pre-exposure showed that B- and T-cells make up a total of 55.21% of the predicted immune cell population. This dropped to the lowest percentage of 7.92% at 7 DPI. There was an increase in predicted monocytes and macrophages starting at 3 DPI and increasing through 8 DPI. Predicted percentage of monocytes and macrophages started at 6.08% at baseline and increased to 46.86% at 8 DPI. An increase of predicted neutrophils occurred from 2 DPI through 5 DPI and began to decrease through 8 DPI, reaching 59.48% at 6 DPI, up from baseline value of 25.69%.

Processing and analysis by RT-qPCR of samples determined both the qualitative scoring of infection state, as well as a semi-quantitative estimate of viral load in multiple matrices. Some whole blood samples showed detectable virus as early as 2 days post-infection, and in all NHPs by 4 DPI, while other matrices had virus at later DPI (Figure 1d). Analysis for the presence of Ebola antigen VP40, GP, and NP yielded positive results by DPI 4 in most plasma NHP samples, however no significant IgM or IgG response was seen in any of the NHP samples (Figure 2 and Appendix A).

### 3.2. Differential Gene Expression Trends

Evaluation across the 754 genes in the NanoString NHP Immunology v. 2 panelcodeset, showed 528 were differentially expressed genes (DEGs) for at least one timepoint during infection (Appendix A). Differential gene expression (DGE) analysis using a combination of the day −7 and day −14 DPI blood draws for pairwise comparisons to post-exposure timepoints produced no differentially expressed genes (DEGs), therefore we combined both sample sets as a pre-exposure baseline. As only one NHP survived to 8 DPI, one sample was used for DGE analysis at this timepoint. Our analysis did not include the control NHP. DGE analysis resulted in significant fold changes for genes starting at 1 DPI. The number of DEGs increased in number and fold change with 2 DPI through 8 DPI (Figure 3a). The greatest numbers of DEGs were seen at 7 DPI; 425 total with 95 downregulated and 320 upregulated, coinciding with peak clinical illness. Of note, five genes differentially expressed at 1 DPI included interferon-stimulated genes *IFIT2*, *IFIT3*, and *IFI44*. Decreased expression of the chemokine *CCL3* at 1 DPI later increased, peaking at 5 DPI. *IL8* is upregulated at 1 DPI and demonstrates fluctuating levels of expression over the course of disease (Figure 3b). These genes appeared prior to the emergence of physical symptoms of illness and prior to detection of virus in whole blood for a majority of animals. Antigen presentation-related genes were strongly impacted over the course of infection with the genes *TAP1*, *TAPBP*, *HLA-A*, *HLA-B*, *B2M*, *MR1*, *PSMB8*, and *PSMB9* showed strong upregulation. In contrast, EVD infection resulted in downregulation of many MHC II-specific genes at multiple timepoints, including *CD74*, *HLA-DMA*, *DLA-DPA1*, *HLA-DPB1*, *HLA-DQA1*, *HLA-DRA*, and *HLA-DRB1* (Appendix A).

Further exploration of mRNA expression changes during EBOV infection with whole blood NHP samples binned based on viremic status coupled with overt symptoms (See Section 2) demonstrated expression patterns specific to phases during disease course. VENN diagrams corresponding to these bins showed most DEGs significantly expressed across mid and late, 183, or early, mid, and late phases, 136, of the disease progression (Figure 3c and Appendix A). Only 4 genes ubiquitously expressed at significant levels across all phases of infection. These included *IRF7*, *S100A8*, *S100A9*, and *IFI44*. *S100A9* encodes the protein complex calprotectin; known to be secreted during inflammation [35]. *IRF7* is a vital interferon regulatory transcription factor. Interestingly, several genes were found significantly expressed in selective phases including 7 genes: *CD19*, *FCGR3*, *PRDM1*, *IRF1*, *CXCR2*, *IL13RA1*, and *CD79A*. These genes were found only in the early stages of infection. Multiple genes, 42 and 80 respectively, were uniquely significantly expressed during the mid and late phases of infection.

Biological significance through pathway analysis of DEGs with Ingenuity Pathway Analysis (IPA) resulted in predicted upstream regulators and canonical pathway scoring. Comparison analysis across all timepoints demonstrated the top 20 canonical pathways ranked by z-score fall into predominantly immune response pathways. Peak pathway engagement in terms of z-score occurred from 3 DPI through 6 DPI, coinciding with the emergence of serious signs of illness (Figure 3d). The two pathways with negative scoring, “LXR/RXR Activation” and “PPAR Signaling,” are involved in the inhibition of immune responses. The most prominent pathways involved immune response pathways including activation of signaling pathways such as Necroptosis, Neuroinflamation, p38 MAPK, IL, and interferon signaling. Comparison analysis for upstream regulators also resulted in many molecules involved in the immune response with lipopolysaccharide and IFNG being the top predicted upstream regulators. Downregulated upstream regulators included *IL1RN*, which inhibits *IL1B*, and *TRIM24*, a transcription factor implicated in the inhibition of interferon signaling [36]. Many of the predicted upstream regulators are present as DEGs in the dataset, including *IL1B*, *IRF7*, *STAT1*, and *SOCS1*.

### 3.3. Differential miRNA Expression Trends

Differential miRNA/picoRNA (piRNA) expression analysis showed 534 miRNAs and 71 piRNAs present in at least one NHP across all DPI. 93 DEMs were found, including 83 miRNAs and 10 piRNAs, to have differential log2-fold-changes >1 at any given time-point (Appendix A). We observed DEMs as early as 2 DPI and the number continuously increased with progression of disease (Figure 4a). Two miRNAs, hsa-miR-331-3p and hsa-miR-338-3p, were significantly different at 2 DPI. Hierarchical clustering of least-square means resulted in three large clusters of DEMs (Figure 4c). Two clusters had miRNA/piRNAs expression profiles that consistently increased or decreased, 52 and 29 respectively, with disease progression. Of these miRNAs one in particular, hsa-miR-122-5p, had a log2 fold change over 7 by 8 DPI. A third cluster included 12 miRNAs that demonstrated transient expression peaking at days 3 and 4 DPI then decreasing with disease progression. These 12 miRNAs represented approximately 70% and 56% of the day 3 and 4 DEMs respectively indicating a transient early miRNA profile during EVD progression.

To further explore this transient expression profile, analysis was performed on samples binned as described above. No DEMs were found with log2-fold changes >1 in the incubation phase, however, two miRNAs, hsa-miR-331-3p and hsa-miR-338-3p, had significant p-value scores with log2-fold changes >0.6 (Appendix A). These two miRNAs were identical to those found to be significant at 2 DPI in the analysis above. A total of 8 DEMs were found in the early phase of disease progression; VENN diagrams revealed 4 of these 8 miRNAs, hsa-miR-1262, hsa-miR-4304, hsa-miR-4784, and hsa-miR-7977, were significantly expressed only in the early phase of disease. Most DEMs in the mid phase were shared with late phase samples with the exception of hsa-piR-020813-gb-DQ598646-Homo and hsa-miR-34a-5p. KEGG pathway analysis was performed using DIANA-miRPath v3 on DEMs found for each binned phase [37]. Of the 8 miRNA/piRNAs found in early samples, 7 miRNAs were used for KEGG pathway analysis. The top three pathways found to be significant using an FDR corrected p-value threshold of 0.05 were signaling pathways including MAPK, ErbB, and Oxytocin. Similarly, for mid and late phase samples, signaling pathways including ErbB, Ras, Hippo, and AMPK, were highly significant (Appendix A).

### 3.4. Correlative Analysis of Significant miRNA and mRNA Signatures

Within our dataset of DEMs, IPA identified 71 unique miRNAs experimentally observed or highly/moderately predicted to regulate 510 mRNAs on our NanoString panel. To determine if miRNA expression correlated with our mRNA data, Spearman Rho correlations were performed using NHP matched log2 normalized counts across all samples. In total 71 miRNA-mRNA pairs had inverse expression relationships with Spearman Rho correlations <−0.5 and a significant probability ≤0.0001 (Table 1). Positive correlations were ignored for this analysis, as this would indicate an indirect method of miRNA regulation of mRNA transcripts. Of these 71 miRNA-mRNA pairs, three had a Spearman Rho correlation <−0.7.

We assessed correlations between mRNAs within our NanoString panel and the seven DEMs, hsa-miR-125b-5p, hsa-miR-1262, hsa-miR-7977, hsa-miR-4784, hsa-miR-338-3p, hsa-miR-144-3p, and hsa-miR-4304, that were identified early in disease progression. Predictive analysis suggested these miRNAs regulated 181 mRNAs within the NanoString panel of which 10 miRNA-mRNA pairs had Spearman Rho correlations <−0.5 and a *p*-value ≤0.0001. Of these 10 pairs, miRNA hsa-miR-125b-5p was over-represented and correlated with mRNAs *IL16*, *ETS1*, *BCL2*, *PPP1R12B*, *IRF4*, *CD244,* and *CASP2*. Interestingly, the early phase miRNAs hsa-miR-144-3p and hsa-miR-338-3p, were also significantly correlated with *ETS1* gene repression (Table 1).

## 4. Discussion

Future efforts in Ebola virus research and clinical treatment depend on improved diagnostics and medical countermeasures. Definition of reliable host biomarkers/classifiers of EVD determinant of disease status, both diagnostically and prognostically, will be invaluable. To this end, we sought to further refine host transcriptomic miRNA and mRNA signatures in a key animal model of EVD. We focused this analysis on identification of host miRNAs involved in EVD pathogenesis. Our focus on miRNAs as biomarker targets of interest continues previous research by our group evaluating these molecules as potentially robust host diagnostic targets for viral hemorrhagic fevers including EVD [38]. We investigated miRNAs occurring during EBOV pathogenesis in a key NHP model of EVD. For all EBOV-challenged cynomolgus macaques in this study, the clinical progression of EVD was consistent with previous reports for this model and was uniformly fatal (Figure 1a, Appendix A). Our RT-qPCR data demonstrated escalating viremia in all challenged NHPs, consistent with fulminate infection and an insufficient host immune response (Figure 1c,d, Figure 3a and Figure 4a). As such, this challenge cohort appears typical of the EVD NHP model, and was suitable for investigation of disease-associated biomarkers.

Severe EVD pathogenesis in primates bears many of the hallmarks of sepsis and is frequently described as having two distinct phases: (1) an initial pro-inflammatory phase characterized by widespread activation of cytokines/chemokines like TNF-α and IL-1β, and (2) a disrupted anti-inflammatory response that can limit successful lymphocyte activation and function, and which is distinguished by dramatic lymphopenia [6]. In severe cases, the end result of this immune dysregulation is impaired multi-organ failure, septic shock, and death [9]. In this study, we detected the pro-inflammatory transcriptional response as early as 1 DPI with full engagement in all NHPs by 3 DPI. Here, we observed some mRNA DEGs, including *CCL3*, *IF44*, *IFIT2*, and *IFIT3* appearing prior to both onset of clinical signs of illness and detection of EBOV genome by qPCR (Figure 1d and Figure 3b). These early mRNA biomarkers primarily belong to the family of interferon-stimulated genes (ISGs) implicated in the host anti-viral responses associated with a number of other RNA viruses [39,40,41,42]. Pronounced ISG upregulation is a known biomarker for severe EVD with higher viremia stimulating higher inflammatory responses. Indeed, in both human and animal models, strong ISG upregulation often correlates with fatal outcomes for EVD [16]. mRNA pathway analysis further highlighted this early pro-inflammatory phase emerging by 1 DPI with gene expression pathways associated with hypercytokinemia (Figure 3d). Interestingly, we identified numerous pro-inflammatory genes that are associated with both interferon signaling and host response to bacterial lipopolysaccharide (LPS) stimulation. In vitro studies have previously reported that both LPS simulation and EBOV infection of target cells trigger many of the same inflammatory signaling responses, including TLR4 activation and ISG upregulation [43]. Additional pathways that showed signs of early and continuous activation based on mRNA expression included *IL6*, *IL17*, and *p38 MAPK* signaling, all of which are associated with a pro-inflammatory state commonly seen in EVD [44,45,46,47]. Beyond activation of innate antiviral immune defenses, this exaggerated inflammatory state, and its associated cytokines also drive premature activation of B and T-cell lymphocyte populations [6,12].

As expected from this fatal disease model, pathway analysis of our NHP data indicated massive dysregulation of numerous immune pathways related to viral infection (Figure 3c). Positive regulation of p38 MAPK signaling in later timepoints was consistent with previous studies showing that p38 signaling is important for EBOV infection of antigen presenting cells (APCs) [46]. The upregulation of “Production of Nitric Oxide and Reactive Oxygen Species in Macrophages” and “iNOS Signaling” pathways are consistent with other studies that showed an elevated level of nitric oxide in the blood of both human and NHP EVD [10]. This increased nitric oxide synthesis contributes to lymphocyte apoptosis and vascular tissue damage, both hallmarks of EVD. During early infection, transcriptional factors associated with negative regulation of immune function, including *IL1RN*, *SOCS1*, and *TRIM24* were downregulated. Suppression of these factors likely contributes to the observed pro-inflammatory state. Later timepoints displayed downregulation of LXR/RXR activation and peroxisome proliferator-activated receptors (PPARs) transcriptional factors, which are important for lipogenesis and other core metabolic homeostatic functions in numerous organ systems, including the liver [10]. Both *PPAR* and *LXR/RXR*, which were also identified in an earlier EBOV NHP transcriptomic study by our group, are crucial for successful anti-inflammatory host regulation. In addition to these pathways, we noted that gene expression associated with antigen presentation was dysregulated during infection similar to previous reports [12,48]. In particular, many MHC-II-related genes were downregulated starting at 4 DPI, including CD74 and numerous HLA markers.

Our integrated analysis of miRNA interactions with mRNA expression during EVD in NHPs revealed potential roles for multiple host miRNAs in contributing to EBOV pathogenesis. Numerous host miRNAs correlated closely with down-regulation of key immunomodulatory genes involved in both innate and adaptive immunity (Table 1). The microRNA hsa-miR-122-5p was highly upregulated in most NHPs by day 5 (Figure 4b,c, Appendix A), and present in all late phase disease NHPs. Detectable in both the blood and liver, hsa-miR-122-5p has been mostly studied as a marker of liver injury for hepatotropic viruses including Hepatitis C [49,50,51]. Some mRNAs highly correlated with hsa-miR-122-5p included the genes *CCL5*, *CD3E*, and *AKT3* (Table 1). *CD3E*, a marker of T-cell development appears targeted for degradation by hsa-miR-122-5p and by hsa-miR-143-3p over the course of EVD in this study [52]. *CCL5* (RANTES), a pro-inflammatory chemokine associated with T-cell activation and recruitment to sites of inflammation is known to be upregulated in human survivors of EVD, even years after infection [53]. Its apparent downregulation in our study suggests a potential miRNA-induced subversion of this defense mechanism during EBOV pathogenesis. A mediator of the innate immune response that appears implicated with both miR-122-5p and hsa-miR-424-5p is the serine/threonine kinase *AKT3*. While *AKT3* is known to be involved in the PI3K signaling pathway and to play a role in the innate immune response, its relevance for EBOV infection is not fully understood [54]. Of note, however, an in vitro siRNA screen of cells infected by EBOV suggested a role for *AKT3* in limiting EBOV infection, suggesting that miRNA downregulation of this target could in fact undermine a host defense mechanism [55]. In the cases of CCL5 and CD3E, these regulatory molecules appear to be dysregulated by these miRNAs during infection, likely contributing further to EVD pathogenesis and fatal outcome. Similarly, expression of the miRNA hsa-miR-424-5p was associated with downregulation of the gene *IL7R*, an important factor for B and T-cell development [56].

Another significant host miRNA in our transcriptomic analysis that appears closely linked to some of the aforementioned immune dysregulation mechanisms is hsa-miR-125b-5p. This miRNA was significantly upregulated in all three phases of viremic infection (Figure 4b,c, Appendix A). Messenger RNAs whose downregulation appear closely associated with its expression include *IL16*, *ETS1*, *BCL2*, *PPP1R12B*, *CASP2*, and *IRF4* (Table 1). Similar to the previously described inhibitory action of hsa-miR-122-5p on T- cell activation, hsa-miR-125b-5p may impair these immune cells as well, in this case through the degradation of the immunomodulatory proteins IL16 and PP1R12B, both of which play a role in successful CD4+ T-cell activation [57,58]. Their inhibition by this miRNA would likely subvert this cell type’s ability to counter EBOV infection. In a similar vein, hsa-miR-125b-5p’s strong association with IRF4 repression in our study also links it with inhibition of B lymphocyte function. Based on the study of B-cell leukemia, hsa-miR-125b-5p is known to repress IRF4 activity and with it the ability of B cells to undergo germinal center differentiation to antibody producing plasma cells [59]. As shown in Figure 2 and Appendix A, despite abundant circulating viral antigens, virtually no EBOV-specific IgM or IgG response was observed in this study; failure to mount a successful humoral response is a hallmark of severe and fatal cases of EVD in humans [6]. Another significant miRNA-mRNA interaction potentially impactful in B and T-cell activation and proliferation was that of hsa-miR-21-5p on the procaspase-encoding gene *MALT1* [60]. Similarly, expression of the miRNA hsa-miR-223-5p was observed to be closely related to downregulation of the Natural Killer cell activating gene *KLRC3*. Natural killer cells are important elements of the immune response to viral infection, in particular the clearance of virus infected cells; their suppression in fatal cases of EVD in humans and NHPs has been observed [61,62]. KLRC3 is a known regulator of NK cell activity and its downregulation is associated with reduced NK cell function [63,64,65].

A notable feature of fatal EBOV infection is the widespread apoptosis of B and T lymphocytes during infection, not as a result of direct infection, but because of pro-apoptotic stimuli directed towards the lymphocytes by their cellular environment, a process commonly called “bystander apoptosis” [6]. It is of particular interest that many of the downregulated mRNAs detailed above, including *ETS1*, *BCL2*, and *CASP2* are involved in apoptosis and are all known to be negatively regulated by hsa-miR-125b-5p expression [66]. ETS1 is an antiviral transcription factor known to positively upregulate *BCL2*, which is itself a known repressor of apoptosis [67,68]. By targeting these proteins for degradation, hsa-miR-125b-5p and other *ETS1* or *BCL2*-targeting miRNAs identified in this study like hsa-miR-424-5p, hsa-miR-144-3p and hsa-miR-338-3p are likely driving lymphocytes further down the apoptotic pathway. Lymphopenia is well documented in both human and NHP infections studies and this dataset is consistent with this phenomenon, showing a drastic decrease in B-cells and T-cells over the course of infection (Figure 1c) [11,62,69,70].

Previous studies reported upregulation of hsa-miR-122-5p during EBOV infection [38,71], however, its role in EVD pathogenesis remains unclear and its association with downregulation of the immune targets described herein during EVD is novel. The identification of this and other significant host miRNAs in our study, in particular hsa-miR-125b-5p, and their possible inhibition of numerous genes involved in the orchestration of a successful anti-EBOV immune response in primates is notable. An outstanding question with regard to the role that these miRNAs play is how much of this selective miRNA expression is a virus-directed process and how much is a downstream consequence of other pathophysiological processes that systemic EBOV infection is precipitating [20,25]. Those miRNAs that we identified that most closely correlate with immune suppression only begin to emerge during the mid to late infection phase, after upregulation of multiple pro-inflammatory mRNAs. Since an exaggerated pro-inflammatory state, such as that seen during the early infection phase in our NHPs, is known to prematurely activate components of the adaptive immune system, one possibility is that this miRNA response may be part of an immune dampening mechanism designed to prevent premature activation or differentiation of T and B lymphocytes, respectively, both by inhibiting their function as well as by initiating apoptosis. Based on the end result of wide-spread lymphopenia and general unchecked viral dissemination, however, it would seem that this response only further undermines host defenses and contributes to the fatal end state. It is also worth noting that the genes targeted by host miRNAs are still being identified, and therefore it is possible that additional miRNA-mRNA interactions may have gone undetected in the present study. Future investigations should also evaluate whether these miRNA-mRNA interactions are relevant for human EVD.

Pre-symptomatic differential expression of miRNAs in whole blood as early as 2 DPI showed the potential of these markers as diagnostic targets, especially in cases where differential expression appeared before EBOV positivity triggered by qPCR (Figure 1b,c and Figure 4a). If these miRNAs prove relevant for human EBOV infection, their diagnostic utility should be evaluated. miRNAs are remarkably robust and survive nuclease degradation, with previous studies documenting detection in various bodily fluids such as blood (serum, plasma), cerebrospinal fluid, breast milk, urine, and saliva [72]. Similar to work previously done by our lab with hsa-miR-122-5p, it will be of particular interest to evaluate these other miRNA markers on human clinical samples of EVD to confirm their diagnostic potential [38]. Transitioning some of these disease biomarkers to a lower cost, shorter-sample-to-answer assay format such as an RT-PCR assay or point-of-care platforms would be a useful next step.

There are multiple use-case scenarios for an EVD miRNA assay. In one, miRNAs could be used to alert clinicians to patients that are likely to develop a severe EBOV disease state with its attendant immune dysregulation. As has been demonstrated by the clinical responses to the COVID-19 pandemic, patient risk stratification can be a powerful tool in informing optimal resource management practices (staffing, therapeutics, etc.) [73,74]. In a second scenario, these miRNAs could be used for monitoring convalescent EVD patients for signs of long-term disease sequelae. Only after the 2013-2016 West African EVD outbreak and subsequent long-term longitudinal monitoring of patients was it discovered that there is a high incidence of long-term pathologies associated with survival post-acute EVD phase. These include musculoskeletal pain, neurocognitive, and ocular disorders. Furthermore, there appear to be long term immunological consequences from EVD including severe immune function abnormalities and a persistent elevated inflammatory profile [53]. The utilization of a miRNA detection approach including such markers described here could be advantageous for identifying subtle long-term modulation of the immune response and shed light on the pathophysiology of this disease in survivors.

## Figures and Tables

**Figure 1 microorganisms-09-00665-f001:**
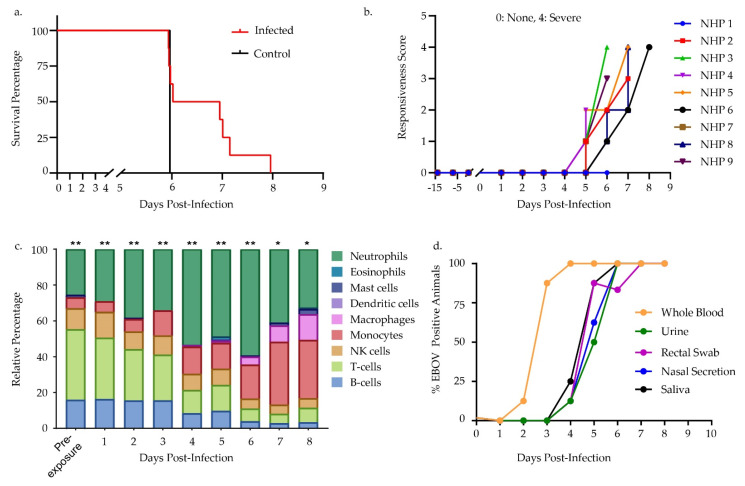
Clinical response, viremia, and digital sorting of whole blood non-human primate (NHP) samples infected with EBOV. (**a**) Kaplan-Meier survival curve of 9 cynomolgus macaques in this study. NHP 1 was a sham-challenge with diluent whereas NHPs 2–9 were challenged intramuscularly with 152 PFU of EBOV. (**b**) Responsiveness scores. Scale of 0–4 with 0 displaying no symptoms and 4 showing severe symptoms. NHP1 data represent sham infected control. (**c**) Digital cell sorting of whole blood transcriptomic data. Relative percentage of immune cells present in whole blood were estimated based on normalized count data and compared to a human control dataset (LM22). P-values were calculated to determine the statistical significance of the deconvolution result across cell types. * indicates *p*-value ≤0.05 and ** indicates *p*-value ≤0.005. All NHPs for each timepoint were averaged prior to analysis save 8 days post-infection (DPI) which only contained one NHP. (**d**) Percentage of samples positive with viral RNA as determined by RT-qPCR in multiple matrices; excluding sham infected NHP #1.

**Figure 2 microorganisms-09-00665-f002:**
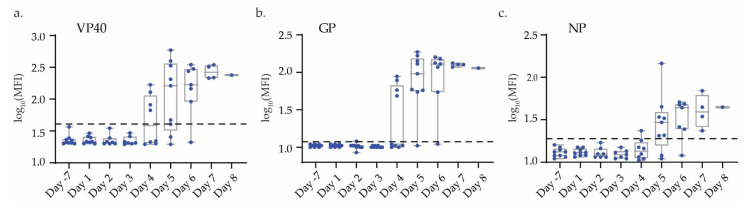
VP40, GP, and NP antigen, response in EBOV infected NHPs. Box and whisker plots for MFI of serum (**a**) viral protein (VP40), (**b**) glycoprotein (GP), and (**c**) nucleoprotein (NP) levels across all timepoints. Whiskers represent min and max values, box indicates the 25th and 75th, and centerline indicates the mean. Dots represent each NHP. The average of uninfected (day 7) samples plus three standard deviations were used as a cutoff and represented by a dashed line.

**Figure 3 microorganisms-09-00665-f003:**
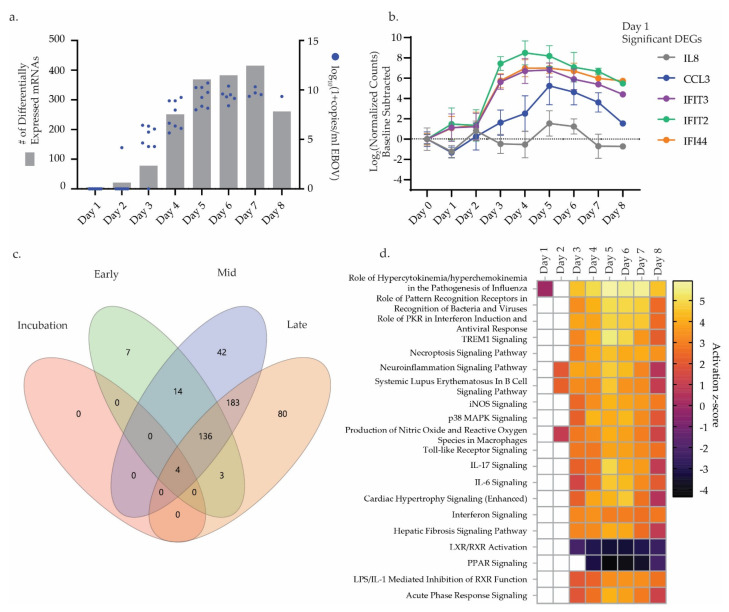
mRNA expression profile of NHPs infected with EBOV. (**a**). Total number of significantly expressed mRNAs in whole blood NHP samples at each DPI plotted against log_10_(1+copies/mL) Ebola virus as determined by quantitative RT-PCR. (**b**). Log2 (normalized counts) of significant differentially expressed genes (DEGs) at 1 DPI plotted versus DPI. Data were baseline subtracted to 0 DPI. (**c**) Samples were binned into for groups according to responsiveness and viremia (see methods). VENN diagram displays overlap of significantly expressed mRNAs across all groups. (**d**) Top 20 significant pathways determined by IPA analysis of DEGs ranked by activation z-score.

**Figure 4 microorganisms-09-00665-f004:**
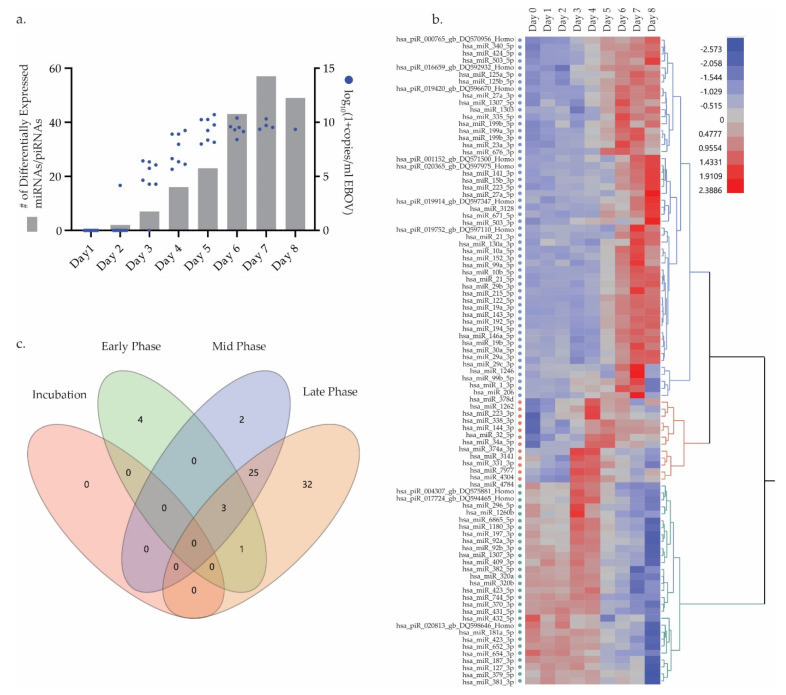
Differentially expressed miRNAs/piRNAs (DEM) expression profile of NHPs infected with EBOV. (**a**) Total number of DEMs at each DPI plotted against log_10_ (1+copies/mL) EBOV as determined by quantitative RT-PCR. (**b**) Hierarchical clustered heat map of least square means calculated for each significantly expressed DEM across all timepoints. Colors indicate 2 clusters of continuously upregulated/downregulated DEMs and 1 cluster of transiently expressed DEM (**c**). VENN diagram of DEMs resulting from binning samples into four groups according to responsiveness and viremia.

**Table 1 microorganisms-09-00665-t001:** Spearman Rho correlations of log2 (normalized data) between miRNA and mRNA transcripts.

miRNA	mRNA	Spearman Rho	Prob>|Rho|	miRNA	mRNA	Spearman Rho	Prob>|Rho|
hsa-miR-125b-5p *	*IL16*	−0.767	<0.0001	hsa-miR-423-5p	*LTF*	−0.5655	<0.0001
hsa-miR-424-5p	*AKT3*	−0.7083	<0.0001	hsa-miR-143-3p	*GZMK*	−0.5647	<0.0001
hsa-miR-125b-5p *	*ETS1*	−0.7004	<0.0001	hsa-miR-424-5p	*ICOS*	−0.5634	<0.0001
hsa-miR-424-5p	*IL7R*	−0.6876	<0.0001	hsa-miR-423-5p	*CHIT1*	−0.5619	<0.0001
hsa-miR-21-5p	*MALT1*	−0.6873	<0.0001	hsa-miR-199b-5p	*ETS1*	−0.5581	<0.0001
hsa-miR-143-3p	*CD3D*	−0.6799	<0.0001	hsa-miR-122-5p	*CD3E*	−0.5548	<0.0001
hsa-miR-143-3p	*MAP4K1*	−0.6787	<0.0001	hsa-miR-423-5p	*TNFRSF13C*	−0.5541	<0.0001
hsa-miR-223-5p	*KLRC3*	−0.6655	<0.0001	hsa-miR-338-3p *	*ETS1*	−0.5505	<0.0001
hsa-miR-122-5p	*LILRB1*	−0.6615	<0.0001	hsa-miR-19b-3p	*RPS6KA5*	−0.549	<0.0001
hsa-miR-125b-5p *	*BCL2*	−0.6473	<0.0001	hsa-miR-34a-5p	*KLRK1*	−0.549	<0.0001
hsa-miR-122-5p	*CCL5*	−0.6421	<0.0001	hsa-miR-424-5p	*HLA-DQA1*	−0.5477	<0.0001
hsa-miR-199a-3p	*PLCB1*	−0.6324	<0.0001	hsa-miR-199b-5p	*GZMB*	−0.5455	<0.0001
hsa-miR-27a-3p	*IRF4*	−0.6323	<0.0001	hsa-miR-223-5p	*HLA-DPB1*	−0.545	<0.0001
hsa-miR-423-5p	*MAFG*	−0.6271	<0.0001	hsa-miR-19b-3p	*MAF*	−0.5431	<0.0001
hsa-miR-424-5p	*BCL2*	−0.6242	<0.0001	hsa-miR-1260b	*TNFRSF13C*	−0.5386	<0.0001
hsa-miR-1260b	*MAFG*	−0.6224	<0.0001	hsa-miR-423-3p	*MAFG*	−0.5381	<0.0001
hsa-miR-370-3p	*SMAD3*	−0.622	<0.0001	hsa-miR-671-5p	*KLRC3*	−0.5373	<0.0001
hsa-miR-424-5p	*CD3E*	−0.6184	<0.0001	hsa-miR-423-5p	*ATG7*	−0.5363	<0.0001
hsa-miR-27a-3p	*HLA-DRA*	−0.6113	<0.0001	hsa-miR-27a-3p	*MEF2C*	−0.535	<0.0001
hsa-miR-1260b	*TNFSF14*	−0.6082	<0.0001	hsa-miR-19b-3p	*GZMK*	−0.5329	<0.0001
hsa-miR-296-5p	*IKBKE*	−0.6064	<0.0001	hsa-miR-143-3p	*MAF*	−0.5326	<0.0001
hsa-miR-199b-5p	*KLRF1*	−0.6012	<0.0001	hsa-miR-125b-5p *	*CASP2*	−0.5308	<0.0001
hsa-miR-144-3p *	*ETS1*	−0.5972	<0.0001	hsa-miR-29c-3p	*DPP4*	−0.529	<0.0001
hsa-miR-34a-5p	*HLA-DQA1*	−0.5925	<0.0001	hsa-miR-432-5p	*RELA*	−0.5284	<0.0001
hsa-miR-23a-3p	*KLRF1*	−0.5905	<0.0001	hsa-miR-125b-5p	*CD244*	−0.5235	<0.0001
hsa-miR-125b-5p *	*PPP1R12B*	−0.5882	<0.0001	hsa-miR-1260b	*CHIT1*	−0.5228	<0.0001
hsa-miR-10b-5p	*CD3G*	−0.5852	<0.0001	hsa-miR-676-3p	*CD3G*	−0.5212	<0.0001
hsa-miR-143-3p	*HLA-DPB1*	−0.5805	<0.0001	hsa-miR-1260b	*DAXX*	−0.5208	<0.0001
hsa-miR-127-3p	*PLAUR*	−0.5763	<0.0001	hsa-miR-143-3p	*BCL2*	−0.5199	<0.0001
hsa-miR-27a-3p	*ICOS*	−0.5752	<0.0001	hsa-miR-1262 *	*ANP32B*	−0.5167	<0.0001
hsa-miR-6865-5p	*THY1*	−0.5751	<0.0001	hsa-miR-423-5p	*TNFRSF14*	−0.5164	<0.0001
hsa-miR-34a-5p	*KLRD1*	−0.5749	<0.0001	hsa-miR-423-5p	*PLA2G6*	−0.5154	<0.0001
hsa-miR-23a-3p	*BCL2*	−0.5747	<0.0001	hsa-miR-296-5p	*CD22*	−0.514	<0.0001
hsa-miR-21-5p	*TGFBI*	−0.5714	<0.0001	hsa-miR-423-5p	*CSF3*	−0.5125	<0.0001
hsa-miR-125b-5p *	*IRF4*	−0.5683	<0.0001	hsa-miR-29c-3p	*TRAF5*	−0.5113	<0.0001
hsa-miR-223-5p	*DPP4*	−0.5675	<0.0001	-	-	-	-

* Indicates has-miRNAs found to be significantly expressed early in disease progression.

## Data Availability

All data presented in this study are available within the article and Appendix A.

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
