# Peer review of "Transcriptomic Analysis Reveals Host miRNAs Correlated with Immune Gene Dysregulation during Fatal Disease Progression in the Ebola Virus Cynomolgus Macaque Disease Model"

_microorganisms, 2021, doi:10.3390/microorganisms9030665_

Round 1
Reviewer 1 Report
In the submitted manuscript, Stefan and colleagues assess mRNA and miRNA changes in Ebola virus infected Cynomolgus macaques through transcriptome analysis. Their data suggests that strong interactions between miRNAs and immune-related genes may be novel biomarkers of Ebola virus disease. Overall, the manuscript is very interesting with some interesting analyses in a relevant in vivo model of Ebola virus disease.
There were some minor concerns that should be addressed:
- Ensure that italics are used properly in regards to taxonomic classifications of family, genus and species (opening sentences of introduction section)
- Introduce EBOV in full based on current ICTV standards (line 37)
- Capitalize "Central" (line 39)
- In regards to "characteristic filovirus symptoms" (line 41), denote infection or disease here and add references for clinical symptoms
- Genes should be capitalized and italicized (line 78)
- Italicize Macaca fascicularis (line 96)
- Italicize genus names (line 138)
- In Figure 3d, is there any way to change the color scheme such that a more obvious difference is present for the activation Z-scores? The color differences between +1 to +4 are somewhat muted
- Do the authors have any hypothesis or comment for why LPS is a top predicted upstream regulator?
Author Response
Manuscript: “Transcriptomic analysis reveals host miRNAs correlated with immune gene dysregulation during fatal disease progression in the Ebola virus cynomolgus macaque disease model”
Authors: Christopher P. Stefan, Catherine E. Arnold, Charles J. Shoemaker, Elizabeth E. Zumbrun, Louis A. Altamura, Christina E. Douglas, Cheryl L. Taylor-Howell, Amanda S. Graham, Korey L. Delp, Candace D. Blancett, Keersten M. Ricks, Scott P. Olschner, Joshua D. Shamblin, Suzanne E. Wollen, Justine M. Zelko, Holly A. Bloomfield, Thomas R. Sprague, Heather L. Esham, and Timothy D. Minogue
To the reviewers of this manuscript, thank you for your kind consideration and helpful comments/edits. In the attached revised word document, we have incorporated the comments you suggested and integrated them as track changes for your review and acceptance. Listed below are point by point responses to the comments provided by the reviewers. As reviewer 2 did not list any comments, the items below are only from reviewer 1:
Point 1: Ensure that italics are used properly in regards to taxonomic classifications of family, genus and species (opening sentences of introduction section)
Response 1: Italicization of all viral family/genus/species has been corrected on lines 34, 36
Point 2: Introduce EBOV in full based on current ICTV standards (line37)
Response 2: On line 37, “EBOV, formerly Ebolavirus Zaire (EBOV)…” has been changed to “Zaire Ebolavirus (EBOV)…” to comply with current ICTV standards for filoviruses.
Point 3: Capitalize "Central" (line 39)
Response 3: This word has been capitalized.
Point 4: In regards to "characteristic filovirus symptoms" (line 41), denote infection or disease here and add references for clinical symptoms
Response 4: This was clarified to refer to EVD (Ebola virus disease)…references for clinical manifestations of EVD were added on line 43.
Point 5: Genes should be capitalized and italicized (line 78)
Response 5: All genes and references to mRNAs associated with genes have been capitalized and italicized throughout the manuscript (lines 77, 78, 402-411, 418-422, 445-447, 499-501, 528, 542-543, 561, 566, 581, 583, 589, 592, 602, 616, 618, 621, 627-631)
Point 6: Italicize Macaca fascicularis (line 96)
Response 6: This species name on line 95 has been italicized.
Point 7: Italicize genus names (line 138)
Response 7: The bacterial genus names on lines 136-137 have been italicized.
Point 8: In Figure 3d, is there any way to change the color scheme such that a more obvious difference is present for the activation Z-scores? The color differences between +1 to +4 are somewhat muted.
Response 8: The large distribution of the activation z-scores in figure 3D made small differences difficult to project using a gradient color scheme. We selected a different color palette to convey the greatest visual differentiation. A new figure panel has been introduced on line 424.
Point 9: Do the authors have any hypothesis or comment for why LPS is a top predicted upstream regulator?
Response 9: In further reading of the literature, we discovered that there appears to be significant overlap between signaling pathways utilized by host cells in response to both LPS stimulation and viral infection. Indeed, there is RNAseq and cytokine protein data that shows significant overlap of genes upregulated in vitro during cell exposure to LPS or infection by EBOV.
We have commented on this in the Discussion text between lines 537-546 by introducing the following (new text underlined) text change:
“Interestingly, we identified numerous pro-inflammatory genes that are associated with both interferon signaling and host response to bacterial lipopolysaccharide (LPS) stimulation. In vitro studies have previously reported that both LPS simulation and EBOV infection of target cells trigger many of the same inflammatory signaling responses, including TLR4 activation and ISG upregulation 43. Additional pathways that showed signs of early and continuous activation based on mRNA expression included IL6, IL17, and p38 MAPK signaling, all of which are associated with a pro-inflammatory state commonly seen in EVD 44-47. Beyond activation of innate antiviral immune defenses, this exaggerated inflammatory state, and its associated cytokines also drive premature activation of B and T-cell lymphocyte populations 6,12.”
Additional Point: We noticed upon final review of the revised document that figure 4 was missing an inset figure legend for the heatmap in 4D. This was added.
Reviewer 2 Report
This is an extensive analysis of transcriptomics and miRNA analysis in an EBV model in macaques. It adds further information to the previous work by the group on the profile of miRNA during Ebola virus infection. The study is well designed and presented. This information has significant pathogenic and potential diagnostic relevance.
Author Response

(The authors gave the same response as above.)
